# Effect of Percutaneous Electric Stimulation with High-Frequency Alternating Currents on the Sensory-Motor System of Healthy Volunteers: A Double-Blind Randomized Controlled Study

**DOI:** 10.3390/jcm11071832

**Published:** 2022-03-25

**Authors:** David Martín-Caro Álvarez, Diego Serrano-Muñoz, Juan José Fernández-Pérez, Julio Gómez-Soriano, Juan Avendaño-Coy

**Affiliations:** Toledo Physiotherapy Research Group (GIFTO), Faculty of Physiotherapy and Nursing, Castilla La Mancha University, 45071 Toledo, Spain; david.martincaro@uclm.es (D.M.-C.Á.); juanjose.fernandez@uclm.es (J.J.F.-P.); julio.soriano@uclm.es (J.G.-S.); juan.avendano@uclm.es (J.A.-C.)

**Keywords:** nerve block, high-frequency, percutaneous electric stimulation, peripheral nerve

## Abstract

Former studies investigated the application of high-frequency alternating currents (HFAC) in humans for blocking the peripheral nervous system. The present trial aims to assess the effect of HFAC on the motor response, somatosensory thresholds, and peripheral nerve conduction when applied percutaneously using frequencies of 10 kHz and 20 kHz in healthy volunteers. A parallel, placebo-controlled, double-blind, randomized clinical trial was conducted. Ultrasound-guided HFAC at 10 kHz and 20 kHz and sham stimulation were delivered to the median nerve of 60 healthy volunteers for 20 min. The main assessed variables were the maximum isometric flexion strength (MFFS) of the index finger, myotonometry, pressure pain threshold (PPT), mechanical detection threshold (MDT), and sensory nerve action potential (SNAP). A decrease in the MFFS is observed immediately postintervention compared to baseline, both in the 10 kHz group (−8.5%; 95% CI −14.9 to −2.1) and the 20 kHz group (−12.0%; 95% CI −18.3 to −5.6). The between-group comparison of changes in MFFS show a greater reduction of −10.8% (95% CI −19.8 to −1.8) immediately postintervention in the 20 kHz compared to the sham stimulation group. The percutaneous stimulation applying 20 kHz HFAC to the median nerve produces a reversible postintervention reduction in strength with no adverse effects.

## 1. Introduction

Preclinical studies in animals have shown that a high-frequency alternating currents (HFAC) stimulation >1 kHz blocked the peripheral nerve system, which quickly reverted without damaging the nerve [1,2,3,4,5,6]. To date, there has been no agreement established regarding the optimal frequency to block nerve conduction in humans. A systematic review by Avendaño et al. [7] about HFAC reported a wide range of frequencies (4 kHz to 30 kHz) that could attain nerve block in animals via implanted electrodes. The minimum frequency to reach nerve block depends on the axon type and nerve width. However, the minimum frequency for currents delivered via implanted electrodes to the median nerve in primates, whose diameter is similar to that of humans, was 20 kHz for a complete nerve block [4]. Clinical trials applying transcutaneous HFAC to the peripheral nervous system showed that 5 kHz currents increased somatosensory thresholds, such as mechanical pain, thermal pain, and tactile sensitivity [8,9], and decreased voluntary strength [9,10]. The application of 10 kHz HFAC also showed the immediate inhibition of motor response and sensory perception [9] that could persist up to 10 min after the stimulation [10,11]. A decrease in voluntary strength has also been observed during the transcutaneous application of 20 kHz HFAC [12].

The main limitation of a transcutaneous current application is the distance between the electrode and the nerve. Bhadra et al. [2] observed that the longer the distance from the electrode to nerve, the greater the intensity required to reach nerve block. Ackermann et al. [13] determined that the optimal distance between the electrode and nerve was 1–2 mm, which is not possible during a transcutaneous current application. No research has been found about percutaneous HFAC stimulation with needles in humans, but studies have been conducted with implanted intrafascicular electrodes placing ring-electrodes around the nerve [14,15,16,17]. Percutaneous application using needles could allow for a reduction in the distance between the electrode and nerve [18] and, therefore, could decrease the current intensity required to reach nerve block. Additionally, percutaneous stimulation can avoid the inconveniences associated with the surgical implantation of electrodes, such as scars or additional procedures to replace them [19]. This intervention has been shown to be safe and with minimum risks in low- [20] and high-frequency [21] electrical stimulations. The main aim of this trial is to assess the effect of percutaneous electric stimulation with 10 kHz and 20 kHz HFAC applied to the median nerve compared to the sham stimulation on the motor response, somatosensory thresholds, and peripheral nerve conduction in healthy volunteers. The secondary objectives are to evaluate adverse effects of the stimulation and subjective perceptions of the participants, proximal and distal temperature changes at the stimulation area, and to assess the blinding success.

## 2. Methods

This study was performed according to the Consolidated Standards of Reporting Trials (CONSORT) standards’ protocol [22]. The present study was approved by the local Toledo Ethical Committee (ref. no 441; 11 November 2019) and the clinical trial was registered in the ClinicalTrials.gov Protocol Registration System (NCT04346719). Participants were informed about the protocol and signed the informed consent approved by the local ethics committee. This study was carried out in the laboratory of GIFTO group at the Castilla-La Mancha University.

### 2.1. Design

A double-blinded, randomized, controlled, parallel clinical trial was designed. Epidat 3.1 software was employed for the simple balanced randomization of the sample. Subjects (*n* = 60) were randomly assigned to the three intervention groups (10 kHz HFAC, 20 kHz HFAC, and sham stimulation) by an independent investigator. Both participants and assessors were blinded to the group assignment, which was kept in a closed envelope throughout the intervention so that only the researcher who delivered the intervention was aware of group allocations. The intervention lasted 20 min. Variables were measured at four time points: (i) preintervention (0 min), (ii) during the intervention,15 min following the start of the stimulation (15 min), (iii) immediately postintervention (20 min), and (iv) 15 min after the finalization of the intervention (35 min) [8]. However, both the antidromic sensitive action potential (SNAP) and maximal flexion finger strength (MFFS) were not recorded during the intervention because of interferences with the application of electric currents and the discomfort needles caused during muscle contractions, respectively.

### 2.2. Subjects

Healthy volunteers were recruited, ranging 18 to 40 years of age, with no pathologies of the nervous system and with no allergy to nickel or intolerance to the percutaneous application of electric currents. The criteria for exclusion were: surgical procedure or osteosynthesis material in the upper limb where the electric stimulation was to be applied, epilepsy, fear of needles, infectious disease, neuro-muscle disease, heart failure, diabetes, cancer, pacemaker or other implanted electric devices, pregnancy, tattoo or skin condition in the area not allowing for the delivery of the intervention, and use of substances or medication (e.g., anticlotting, thrombolytic, analgesic, corticoid, antidepressant, antiepileptic) during the trial and in the seven days before their participation.

### 2.3. Intervention

The duration of both active HFAC and sham stimulation was 20 min. The interventions were delivered to the nondominant arm with the participants in the supine position. Antiseptic and skin disinfection treatment with 2% chlorhexidine in an alcohol base was applied on the intervention area. A Samsung HS50 (Samsung healthcare; Seoul, South Korea) ultrasound device was employed for the percutaneous guided application of currents with a lineal probe of 12 mHz. A short-axis approach to the median nerve was performed on the anterior aspect of the middle third of the forearm placing two 0.30 mm × 40 mm acupuncture needles (Agupunt^®^; Barcelona, Spain) close to the epineurium of the median nerve (1 mm), one needle on each side of the nerve. The average depth of the needle introduced into the tissue was 3 cm (Figure 1).

A Myomed 932 (Enraf-Nonius; Delft, The Netherlands) device connected to the needles by a clamp applied the current in the three interventions. All interventions were delivered in a university laboratory facility under reduced noise and temperature (21 °C–25 °C) conditions. High-frequency alternating currents (HFAC) with sinusoidal waveform were applied at a frequency of either 10 kHz or 20 kHz in the active intervention groups. The intensity progressively increased until producing a feeling of “strong but comfortable” tingling, just under the motor threshold. Subsequently, the intensity was increased until a minimal visible contraction was observed and then slightly lowered below the motor threshold. Due to the accommodation to the stimulus, the intensity was adjusted every two minutes to rise it if the participant’s perception of the current decreased [23]. Sham stimulation was applied with the same device and needle placement. The same parameters were used as in the 10 kHz intervention except for the current intensity, which was initially adjusted up to the sensitive threshold and, once the participant perceived a tingling sensation for a few seconds, the intensity was gradually lowered to and maintained at 0 mA for the entire session.

### 2.4. Outcome Measures

#### 2.4.1. Maximum Isometric Finger Flexion Strength and Myotonometry

The main outcome measurements related to motor activity were the MFFS of the index finger and mechanical characteristics of the opponens pollicis muscle as assessed through myotonometry. MFFS of the index finger was evaluated with the participant in the supine position and their hand in pronation pressing on a MicroFet 2TM digital hand dynamometer (Hoggan Scientific, LLC; Salt Lake City, UT, USA), a device with proven intra- and inter-assessor reliability [24]. The MFFS was calculated as the mean of three measurements in kgs that were taken with a contraction time of 3 s and a rest of 5 s between measurements [25]. A myotonometer (Myoton AS; Tallinn, Estonia) was used to evaluate the mechanical properties of the muscle. Ten mechanical stimuli of 0.4 N force and 0.15 ms duration were applied to the opponens pollicis muscle of the limb where the intervention was performed with one-second intervals between stimuli. If the variation coefficient exceeded 3%, the measurement was repeated. Stiffness (N/m), frequency (Hz), and logarithmic decrement (expressed in arbitrary units) were the collected variables. Stiffness measured the force that resulted in tissue shape changes. The frequency of damped oscillations served to measure the resistance of the tissue to mechanical stress and was considered to be an indirect measure of muscle tone. The decrement served to characterize tissue elasticity by measuring the loss of mechanical energy as the amplitude of the oscillations decreased [26,27].

#### 2.4.2. Mechanical Detection Threshold and Pressure Pain Threshold

Somatic sensitivity was evaluated by means of the mechanical detection threshold (MDT) and pressure pain threshold (PPT). The MDT was measured via modified Von Frey filaments (OptiHair2, MARSTOCKnervtest; Marburg, Germany) on the palmar aspect of the hand in an area of 1 cm^2^ proximal to the head of the second metacarpal and on the thenar eminence. Filaments with a diameter of 0.4 mm delivered forces of 0.25, 0.5, 1, 2, 4, 8, 16, 32, 64, 128, 256, and 512 mN [28]. Seven stimuli were applied, and the threshold was determined when at least four were perceived with a filament [29]. The PPT was recorded on the palmar aspect of the trapeziometacarpal joint via a digital algometer with an increment scale of 0.1 N (Wagner Instruments, model FDIX; Greenwich, CT, USA) and a circular applicator of 1 cm in diameter. The pressure was increased at an approximate rate of 5 N/s [30]. Three measurements were taken with an interval of 10 s between them [31] and the PPT (N) was obtained from the average of the three measurements [32,33,34].

#### 2.4.3. Antidromic Sensory Nerve Action Potential (SNAP)

The SNAP of the median nerve was recorded for assessing the effect on peripheral nerve conduction [35]. Nerve stimulation was performed on the inner side of the arm using a transcutaneous bipolar electrode, with a fixed distance between electrodes of 1 cm and placing the cathode 40 cm from the recording electrode. Two ring electrodes on the index finger were employed to record the potential, with the ground electrode placed on the radial side of the wrist joint [36,37]. A constant-current stimulator (Digitimer LTD, model DS7A; Letchworth Garden, UK), an analogic/digital data acquisition card (Cambridge Electronic Devices; Cambridge, United Kingdom), and an amplifier (ETH-256 iWorxs; Dover, DE, USA) with a 3 Hz high-pass filter and a 2000 Hz low-pass filter and an amplification of 1 were employed for the stimulation and recording. Supramaximal stimuli were applied with a pulse width of 1000 µs and a frequency of 1 Hz. The latency and amplitude of the SNAP were calculated as the mean value of ten measurements. At baseline, two SNAPs were recorded with a 2 min interval to analyze the power stability, and the average was used as the basal SNAP value.

#### 2.4.4. Temperature of Forearm and Arm

A temperature monitor (model DRT4, Moor Instruments brand; Devon, UK) was employed to record temperature. One recording sensor was applied distal to the procedure on the palmar side of the head of the first metacarpal, and another sensor proximal to the procedure on the anterior side of the forearm [8]. The room ambient temperature was also recorded.

#### 2.4.5. Adverse Effects and Subjective Perception

For the evaluation of adverse effects and the subjective perception of the participants, a standardized questionnaire was designed and completed at the end of the intervention. The questionnaire included nine items with “Yes/No” response options to evaluate pain, swelling, heat, redness, coldness, numbness, loss of strength, heaviness, and tingling in the hand and the intervention area. The unpleasantness and pain feeling perceived during the intervention were also assessed using a numerical scale from 0 to 10, where 0 corresponded to “not at all” and 10 to “the maximum possible”. Additionally, participants were asked to report whether they perceived any of the above-mentioned effects or sensations in the intervention area in the days following the intervention.

### 2.5. Assessment of Blinding Success

The blinding success of the participants and the evaluator was assessed after the intervention ended [38]. For this purpose, they were asked “What type of treatment do you believe you or the participant have received?” with five response options: (1) “I strongly believe that I have received an experimental treatment”; (2) “I somewhat believe that I have received an experimental treatment”; (3) “I strongly believe that I have received a placebo”; (4) “I somewhat believe that I have received a placebo”; (5) “Do not know, no answer”.

### 2.6. Statistical Analysis

The sample size was calculated based on a previous pilot test carried out on seven healthy volunteers [21]. For an expected between-group mean difference (MD) in the PPT of 10.3 N/cm^2^ with a standard deviation (SD) of 11.3 N/cm^2^ in the experimental group and SD 9.9 N/cm^2^ in the control group and considering a type I error (α) of 0.05 and a power of 80%, the sample size was estimated to be 17 subjects per group (*n* = 17). To compensate for possible dropouts, a supplementary 17% was added to the sample finally yielding a total of *n* = 20 participants per group. For the comparison of basal characteristics between groups, a descriptive analysis and inferential statistics for basal demographic variables were performed for independent groups (parametric or nonparametric depending on the variable). A two-factor (intervention-time) repeated-measures analysis of variance (ANOVA) with a Bonferroni post hoc was conducted for the following outcome variables: MFFS, myotonometry, PPT, temperature, and SNAP. For those variables violating sphericity, the Greenhouse–Geisser correction was employed. Additionally, changes in the above-mentioned variables over time were calculated and an intergroup comparison was performed via a one-factor (intervention) analysis of variance (ANOVA) with a Bonferroni post hoc. The Friedman test was employed for assessing the MDT with a post hoc analysis via the Tukey’s test for intragroup comparison. The Kruskal–Wallis test was used for the comparison of MDTs between interventions. The Chi-squared test was used for the analysis of adverse effects. Unpleasantness and pain during the intervention were evaluated by means of a one-factor (intervention) ANOVA with a Bonferroni post hoc. All outcome variables were normalized in percentages with respect to basal values prior to the analyses. In the post hoc analysis significant *p*-values are shown at the nominal level alpha < 0.05. The IBM SPSS Statistic 24.0 software for Mac was used for all statistical analyses.

## 3. Results

All sixty randomized participants (*n* = 20 in the 10 kHz group, *n* = 20 in the 20 kHz group, and *n* = 20 in the sham group) completed the trial and were included in the statistical analyses (Figure 2). Table 1 shows the demographic characteristics of the participants. No intergroup differences in demographic variables were found at baseline. The applied current intensity was higher in the 20 kHz group (3.7 mA; SD 2.3) than in the 10 kHz group (1.8 mA; SD 1.3) (*p* = 0.03) at the beginning of the intervention, and also at the end of the intervention, the intensity was higher in the 20 kHz group (12 mA; SD 5.3) compared to the 10 kHz group (6 mA; SD 3.8) (*p* < 0.001). The mean current density was 42.8 mA/cm^2^ and 21.4 mA/cm^2^ for the 20 kHz and 10 kHz groups, respectively. The raw results for all variables could be found in the supplementary material (Appendix A).

### 3.1. Maximum Isometric Finger Flexion Strength and Myotonometry

Table 2 shows the outcomes of MFFS and myotonometry in the study groups across the intervention. Significant differences in MFFS values were observed in the time factor (F = 17.2; *p* < 0.001) but not in the time-intervention intersection (F = 2.4; *p* = 0.07). The MFFS decreased both in the 10 kHz and 20 kHz groups immediately postintervention and 15 min after its finalization versus baseline, in contrast to the sham group where no changes were observed. In terms of the myotonometry results, changes in the frequency were observed in the time factor (F = 7.6; *p* < 0.001) and the time-intervention intersection (F = 2.4; *p* = 0.03). With respect to the baseline, the frequency increased in the 10 kHz group during the intervention and in the 20 kHz group both immediately and at 15 min postintervention. However, no changes were recorded in the sham group. Differences in the decrement parameter were observed in the time factor (F = 4.8; *p* = 0.003) but not in the time-intervention section (F = 1.2; *p* = 0.29). The decrement increased in the 10 kHz group during the intervention and 15 min after its finalization versus baseline, with no changes in the 20 kHz and sham groups. Changes in stiffness measured with myotonometry were noted in the time factor (F = 14.7; *p* < 0.001) and the time-intervention intersection (F = 4.0; *p* = 0.001). Stiffness increased with respect to baseline in the 10 kHz group during and immediately after the intervention and in the 20 kHz group immediately and at 15 min postintervention, in contrast to the sham group where no changes were observed.

Table 3 shows the intergroup comparison of the intervention effect on these variables. Between-group differences in strength were observed immediately postintervention (F = 4.6; *p* = 0.01), specifically a greater strength loss of −10.8% (CI95% −19.8 to −1.8) in the 20 kHz compared to the sham group. No other intergroup differences were reported. Between-group differences in the frequency measured with myotonometry were observed immediately postintervention (F = 3.6; *p* = 0.03), specifically an increase in the frequency in the 20 kHz group versus the sham group. However, no intergroup significant differences in the effect on the decrement and stiffness were observed.

### 3.2. Pressure Pain Threshold and Mechanical Detection Threshold

No differences were found in the PPT in the time factor (F = 2.4; *p* = 0.08) (Table 2) or in the time-intervention intersection (F = 0.5; *p* = 0.78). Additionally, no differences were observed in the intergroup comparison of the effect on the PPT (Table 3). The MDT changed over time (Friedman test *p* = 0.003), but without reaching statistically significant differences in the post hoc analysis (Tukey’s test *p* > 0.05; mean ranks: preintervention = 2.3 mN; during the intervention after 15 min = 2.7 mN; immediately postintervention = 2.5 mN; at 15 min postintervention = 2.5 mN). No intergroup differences were observed at any time point (Kruskal–Wallis test *p* > 0.05).

### 3.3. Antidromic Sensory Nerve Action Potential (SNAP) and Temperature

No significant differences in the median nerve conduction speed were observed in the time factor (F = 2.4; *p* = 0.11) or in the time-intervention intersection (F = 0.6; *p* = 0.60). Significant differences in the potential amplitude were recorded in the time factor (F = 4.8; *p* = 0.01) but not in the post hoc comparison or in the time-intervention intersection (F = 0.3; *p* = 0.88) (Table 2). No intergroup differences were observed either in the comparison of the intervention effect on both the conduction speed and potential amplitude (Table 3). Significant changes in the hand temperature were noted in the time factor (F = 6.9; *p* = 0.002) but without differences in the post hoc analysis. No changes were recorded in the time-intervention intersection (F = 0.3; *p* = 0.87) (Table 2). In terms of forearm temperature, no differences were found in the time factor (F = 0.2; *p* = 0.82) or the time-intervention intersection (F = 1.5; *p* = 0.21) (Table 2). Similarly, no intergroup differences were observed in the comparison of the hand and forearm temperatures (Table 3).

### 3.4. Subjective Variables and Adverse Effects

Table 4 shows the outcomes of subjective variables. Significant intergroup differences were only found in the tingling sensation, and statistical significance was nearly reached in the sensation of strength loss (χ²: 5.71; *p* = 0.057). An unexpected adverse effect was only recorded for one participant in the sham stimulation group. The subject reported a feeling of pain in the forearm and towards the hand that disappeared 48 h after the punction.

### 3.5. Blinding Assessment

Table 5 shows the blinding assessment outcomes for the assessor and participants. The analysis of global blinding, as measured via James’ index [39], yielded the successful blinding of participants and the lack of blinding of the assessor. In the blinding assessment by groups (active and sham) using Bang’s index [38,40], a lack of blinding of both the participants and evaluator in the active group was observed, with 80% and 52% of correctly guessing the group allocation, respectively. In the sham group, 72% of participants thought they were assigned to the active group (opposite guess) and the assessor showed a lack of blinding by guessing the group allocation in 47% of cases.

## 4. Discussion

This was the first clinical trial delivering percutaneous ultrasound-guided HFAC at frequencies of 10 kHz and 20 kHz to the median nerve of healthy volunteers. Percutaneous stimulation at a frequency of 20 kHz showed a significant reduction of 10.8% in MFFS and an increase of 6.7% in the frequency during myotonometry, immediately postintervention when compared to the sham stimulation. Although intragroup changes from baseline were found in the 10 kHz intervention, no differences were found when comparing the 10 kHz to the sham group. Preclinical studies [5,41] suggest that HFAC produces a selective blockade of certain nerve fibers, which can be partial and quickly reversible depending on the frequency and intensity of the delivered current [17]. Nerve fibers react in different ways to nerve blockade depending on the conduction speed [42]. The block threshold varies for each fiber type and also as a function of the stimulation frequency [2,43]. The subjective perception of tingling and strength loss that the present study observed were in agreement with the objective measure of the strength, myotonometry, and MDT variables. The reported adverse effects, such as a cold feeling, postpuncture pain, or heaviness in the area, were light and appeared equally in both active groups, so the percutaneous application of a single session of HFAC could be considered a safe technique with minimum associated risks as well as less invasive and with fewer complications than interventions with implanted electrodes that occasionally require repeated interventions for electrode replacement or result in scarring around the implantation area [19]. However, it is necessary to know the impact of this procedure in repeated applications. It is possible that this procedure could cause greater discomfort to the patient when repeated compared to a single intervention. No participant in the present trial reported a feeling of heat or temperature increase in the application area during the HFAC stimulation, contrary to the studies by Zannou et al. [44,45] who observed a temperature increase in the tissues surrounding the electrodes during the application of 10 kHz currents on the spinal cord simulated by implanted electrodes.

No changes were found in the proximal temperature, in contrast to the decrease in the distal temperature observed in all groups, which could be explained by the percutaneous application of needles. Animal studies have shown that body temperature affects nerve conduction [46,47]. A similar effect on temperature observed in all the intervention groups appeared to indicate a lack of effect on the autonomic nervous system. To date, the selective blocking of the autonomic nervous system [48] has been shown only with the application of toxins in animals. Future research should assess the effect of HFAC on the autonomic nervous system by including specific variables, such as skin flow measured with Doppler laser. This study observed changes in the motor function but not in the sensory function of the nerve, which could be the result of a specific effect of HFAC on conduction in myelinated Aα-fibers. The decrease in the MFFS obtained with the 20 kHz currents occurred immediately after the application ended and the effect persisted up to 15 min. These findings were in agreement with those by Springer et al. [11], who delivered transcutaneous HFCA to the cubital nerve and evidenced an effect that persisted up to 10 min after stimulation. Similar to the outcomes of the present study, Kim et al. [9] and Serrano et al. [10,12] reported a reduction in strength with the transcutaneous application of currents, although this decrease was greater during the application of HFCA. The current study could not assess the effect on strength during the percutaneous application of currents due to the above-mentioned methodological reasons. Further research should delve into the effects on motor fibers during the percutaneous application of currents. In the present study, an increase in myotonometry was found when delivering a frequency at 20 kHz. However, based on the positive correlation between muscle strength and stiffness observed in previous studies [27,49,50], our initial hypothesis was that nerve block would reduce the tone and stiffness, as well as voluntary strength. However, the reduction in muscle strength and the increase in tone and stiffness could be related to a minimal increase in the “noneffective” basal contraction that occurs due to current stimulation. Although the intensity was adjusted below the motor threshold, a small subthreshold contraction could occur that does not interfere with the real nerve block and the decrease in muscle strength. The real impact of this finding should be evaluated in patients with alterations in muscle tone to determine if an amelioration of clinical symptoms such as spasticity, clonus, or tremors can be observed.

The present study did not observe intergroup differences in somatosensory thresholds (PPT dependent on Aδ-fibers and MDT dependent on Aβ-fibers). This was in contrast with the findings of Avendaño et al. [8], who applied 5 KHz transcutaneous HFAC and reported changes in both the PPT and MDT, although these could stem from mechanisms that were unrelated to those involved in nerve blocking [36]. The results in the 20 kHz intervention in the present study were similar to those obtained by Serrano et al. [12] applying transcutaneous HFAC. Dissimilar to the results of the current trial, which did not find an effect on the PPT or MDT, Kim et al. [9] delivered 10 kHz currents and determined that the PPT increased as a function of current frequency. The SNAPs, which are dependent on Aβ-fibers, also did not show detectable changes in the potential amplitude or the conduction speed, unlike the study by Avendaño et al. [8] that observed changes in these variables with the application of 5 kHz HFCA currents. This could be due to the lower frequency that Aβ-fibers need to reach nerve block [5,41,42], although the mechanisms underlying this effect are not clear yet [51]. Although the intensities applied in this work were lower than in transcutaneous application, the values of current density were much higher than in the trials by Serrano et al. [10,12] and Avendaño et al. [8]. As the current frequency increases, a higher intensity is required to reach the nerve block threshold [2,43]. Further studies applying higher frequencies are warranted to determine the intensity needed to attain the nerve block of different nerve fibers without causing damage to the nerve. Some studies have shown that the effect on nerve conduction is reversible, with a ~10 min recovery time [10,11,17,41]. The present trial found that the change observed immediately postintervention in the 20 kHz group compared to the sham group could not persist at 15 min after ending the intervention with HFAC, being consistent with previous studies. The protocol followed in this work could have significant clinical potential in pathologies involving the hyperactivity of the second motor neuron, such as spasticity, tremors, or hypertonia, due to the duration of the effect on motor fibers. Future research on the selective effect of HFAC on nerve conduction is of interest in order to determine the optimal frequency and intensity resulting in a greater effect on the sensory and motor function of nerves.

Given the methodological limitations inherent to the percutaneous application of currents, neurophysiological variables and MFFS could not be measured during the intervention and, therefore, the effect of the intervention on these variables could not be analyzed. The blinding assessment revealed a lack of blinding of the assessor, both globally and by groups, which could have resulted in a detection bias. Future studies with sham stimulation enabling the blinding of assessors are warranted. Another limitation was that the sample comprised exclusively healthy volunteers, so translating these outcomes to clinical practice must be performed with caution until further research in patients is conducted determining the real therapeutic impact of HFAC.

## 5. Conclusions

The percutaneous ultrasound-guided application of HFAC at a frequency of 20 kHz to the median nerve produced a postintervention reduction in strength and an increase in the myotonometry frequency when compared to the sham group, which were rapidly reversible. However, no differences were found between the 10 kHz and sham group. The percutaneous application of HFAC is a safe procedure with minimum associated risks, displaying great potential for treating pathologies affecting the motor function of the nerve without substantial changes in the sensory pathways.

## Figures and Tables

**Figure 1 jcm-11-01832-f001:**
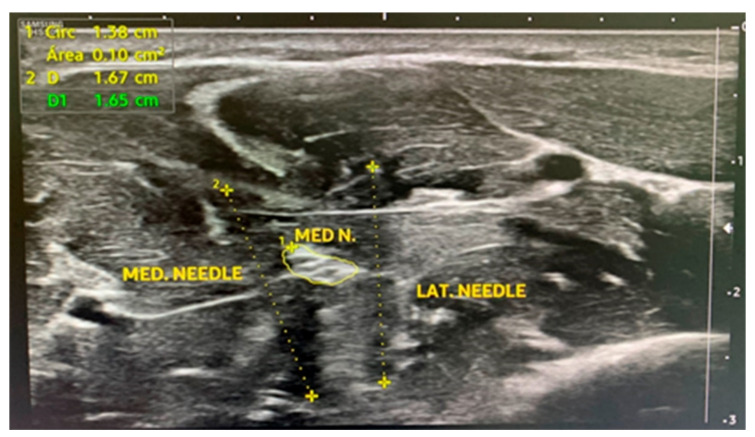
A short-axis approach to the median nerve with two acupuncture needles. MED N.: median nerve; MED NEEDLE: medial needle; LAT NEEDLE: lateral needle.

**Figure 2 jcm-11-01832-f002:**
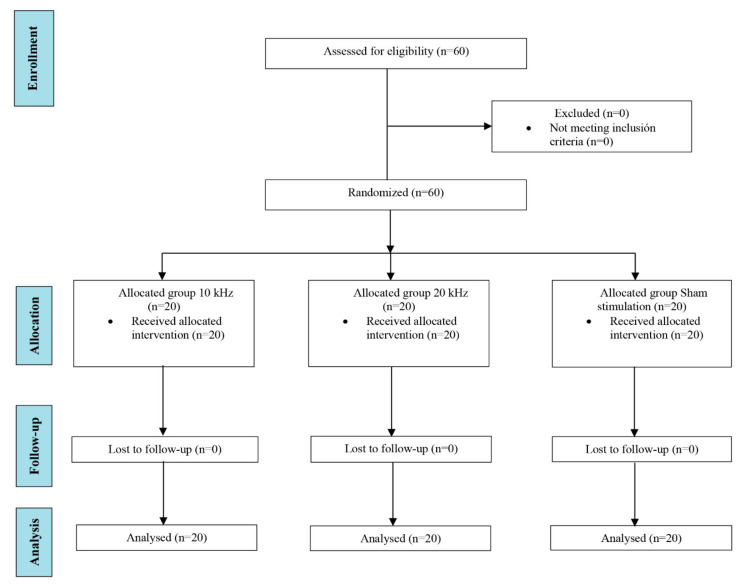
Flowchart showing the process and stages of a parallel randomized clinical trial following CONSORT guidelines.

**Table 1 jcm-11-01832-t001:** Demographic characteristics of participants at baseline. Statistical test: (^a^) one-way analysis of variance (one-way ANOVA), (^b^) Pearson’s chi-squared test, (^c^) Fisher’s exact test.

Outcomes	All Participants (*n* = 60)	10 kHz Group (*n* = 20)	20 kHz Group (*n* = 20)	Sham Group (*n* = 20)	Between Groups Differences (*p* Value)
Age (years) Mean (SD)	21.6 (3.4)	22.9 (4.8)	20.8 (1.9)	21.0 (2.5)	(*p* = 0.09) ^a^
Gender (Male) *n* (%)	26 (43.3%)	11(45.0%)	9(55.0%)	6 (30.0 %)	(*p* = 0.28) ^b^
Weight (kg) Mean (SD)	67.5 (13.2)	71.7 (15.7)	65.2 (11.6)	65.5 (11.6)	(*p* = 0.21) ^a^
Height (m) Mean (SD)	1.70 (0.09)	1.74 (0.10)	1.69 (0.07)	1.68 (0.09)	(*p* = 0.09) ^a^
Body *mass index*(kg/m^2^) Mean (SD)	23.2 (3.2)	23.7 (3.9)	22.6 (3.3)	23.2 (2.4)	(*p* = 0.56) ^a^
Nondominant hand (Left) *n* (%)	57 (95.0%)	19(95.0%)	20(100.0%)	18 (90.0 %)	(*p* = 0.77) ^c^

**Table 2 jcm-11-01832-t002:** Intragroup comparison from baseline. Bold values denote statistical significance. Abbreviations: PRE: preintervention; NA: not applicable; SNAP: sensory nerve action potential.

Outcomes	Sham Group			10 kHz Group			20 kHz Group		
% Mean (CI95%)	During Minus Pre	Post 0 Minus Pre	Post 15 Minus Pre	During Minus Pre	Post 0 Minus Pre	Post 15 Minus Pre	During Minus Pre	Post 0 Minus Pre	Post 15 Minus Pre
Strength	NA	−1.1 (−7.5 to 5.2)	−3.5 (−11.3 to 4.2)	NA	**−8.5** **(−14.9 to −2.1)** ** *p = 0.002* **	**−9.5** **(−17.3 to −1.8)** ** *p = 0.004* **	NA	**−12.0** **(−18.3 to −5.6)** ** *p < 0.001* **	**−11.5** **(−9.3 to −3.8)** ** *p = 0.002* **
Myotonometry Frequency	1.7 (−3.7 to 7.0)	0.4 (−4.4 to 5.1)	1.6 (−2.7 to 6.0)	**6.8** **(1.3 to 12.3)** ** *p = 0.009* **	3.8 (−1.1 to 8.6)	3.1 (−1.4 to 7.5)	4.3 (−1.4 to 10.0)	**7.1** **(2.1 to 12.1)** ** *p = 0.002* **	**6.7** **(2.1 to 11.3)** ** *p = 0.001* **
Myotonometry Decrement	−1.6 (−11.5 to 8.2)	0.8 (−7.9 to 9.6)	3.6 (−4.7 to 12.0)	**10.1** **(0.1 to 20.2)** ** *p = 0.0049* **	7.5 (−1.4 to 16.5)	**11.0** **(2.4 to 19.5)** ** *p = 0.006* **	2.4 (−8.0 to 12.7)	3.9 (−5.4 to 13.1)	6.3 (−2.5 to 15.1)
Myotonometry Stiffness	4.7 (−1.1 to 10.4)	2.6 (−2.5 to 7.8)	2.5 (−2.5 to 7.4)	**10.7** **(4.8 to 16.6)** ** *p < 0.001* **	**6.3** **(1.0 to 11.6)** ** *p = 0.011* **	2.4 (−2.7 to 7.5)	4.7 (−1.4 to 10.7)	**8.5** **(3.1 to 13.9)** ** *p < 0.001* **	**8.0** **(2.8 to 13.2)** ** *p < 0.001* **
Pain Pressure Threshold	−6.5 (−20.2 to 8.1)	−6.2 (−17.9 to 5.5)	−7.4 (−20.1 to 5.3)	−7.9 (−22.1 to 6.2)	−0.7 (−12.4 to 11.0)	−2.6 (−15.3 to 10.0)	−4.6 (−18.8 to 9.5)	−5.6 (−17.2 to 6.1)	−4.3 (−16.9 to 8.4)
Amplitude SNAP	NA	29.0 (−9.1 to 67.2)	26.5 (−7.5 to 60.5)	NA	21.8 (−16.4 to 59.9)	20.9 (−13.1 to 54.9)	NA	31.2 (−6.9 to 69.3)	10.1 (−23.9 to 44.1)
Nerve Speed Conduction	NA	−5.5 (−14.2 to 3.3)	−6.1 (−15.1 to 2.9)	NA	−0.1 (−8.9 to 8.7)	−0.4 (−9.4 to 8.6)	NA	−2.0 (−10.8 to 6.7)	−5.5 (−14.5 to 3.5)
Forearm Temperature	0.8 (−2.3 to 3.8)	0.7 (−1.8 to 3.3)	0.8 (−2.2 to 3.9)	−1.6 (−4.6 to 1.5)	−1.4 (−3.9 to 1.2)	−2.1 (−5.2 to 1.0)	1.2 (−1.8 to 4.3)	0.9 (−1.7 to 3.4)	0.6 (−2.5 to 3.7)
Hand Temperature	−2.4 (−7.5 to 2.7)	−2.7 (−7.1 to 1.6)	−4.1 (−9.2 to 1.0)	−3.1 (−8.2 to 2.0)	−2.1 (−6.4 to 2.2)	−3.3 (−8.5 to 1.8)	−1.4 (−6.5 to 3.7)	−2.3 (−6.7 to 2.0)	−4.1 (−9.3 to 1.0)

**Table 3 jcm-11-01832-t003:** Intergroup comparison in the change from baseline. Bold values denote statistical significance. Abbreviations: NA: not applicable; SNAP: sensory nerve action potential.

Outcomes	Change Sham minus Change 10 kHz			Change Sham minus Change 20 kHz			Change 10 kHz minus Change 20 kHz		
% Mean (CI95%)	During Intervention	Post 0 min	Post 15 min	During Intervention	Post 0 min	Post 15 min	During Intervention	Post 0 min	Post 15 min
Strength	NA	−7.3 (−16.4 to 1.7)	−6.0 (−17.0 to 5.0)	NA	**−10.8** **(−19.8 to −1.8)** ** *p = 0.01* **	−8.0 (−19.0 to 3.0)	NA	−3.5 (−12.5 to 5.5)	−2.0 (−13.0 to 8.9)
Myotonometry Frequency	5.1 (−1.8 to 12.1)	3.4 (−2.7 to 9.5)	1.5 (−4.1 to 7.1)	2.6 (−4.4 to 9.7)	**6.7** **(0.5 to 12.9)** ** *p = 0.03* **	5.1 (−0.6 to 10.8)	−2.5 (−9.6 to 4.7)	3.3 (−2.9 to 9.6)	3.6 (−2.1 to 9.4)
Myotonometry Decrement	11.8 (−1.0 to 24.5)	6.7 (4.6 to 18.0)	7.3 (−3.5 to 18.2)	4.0 (−8.9 to 16.9)	3.0 (−8.4 to 14.5)	2.7 (−8.3 to 13.6)	−7.7 (−20.8 to 5.3)	−3.7 (−15.3 to 7.9)	−4.7 (−15.8 to 6.4)
Myotonometry Stiffness	6.1 (−1.4 to 13.5)	3.7 (−3.0 to 10.3)	−0.1 (−6.5 to 6.3)	−0.02 (−7.5 to 7.5)	5.8 (−0.9 to 12.5)	5.5 (−0.9 to 12.0)	−6.0 (−13.7 to 1.6)	2.2 (−4.6 to 9.0)	5.6 (−0.9 to 12.2)
Pressure Pain Threshold	−1.9 (−20.0 to 16.2)	5.5 (−9.4 to 20.4)	7.0 (−9.3 to 23.3)	1.4 (16.6 to 19.5)	0.7 (−16.6 to 14.3)	9.7 (−6.6 to 26.0)	3.3 (−14.7 to 21.4)	−4.9 (−19.8 to 10.5)	2.7 (−13.6 to 19.0)
Amplitude SNAP	NA	−7.3 (−61.2 to 46.7)	−5.5 (−53.7 to 42.6)	NA	2.1 (−51.8 to 56.1)	−16.4 (−64.5 to 31.7)	NA	9.4 (−44.5 to 63.3)	−10.9 (−59.0 to 37.3)
Nerve Speed Conduction	NA	5.4 (−7.0 to 17.8)	−3.6 (−81.1 to 73.9)	NA	−3.4 (−15.8 to 9.0)	−16.6 (−94.1 to 61.0)	NA	−1.9 (−14.3 to 10.5)	−13.0 (−90.5 to 64.6)
Forearm Temperature	−2.3 (−6.2 to 1.5)	−2.7 (−5.3 to 1.2)	−2.9 (−6.9 to 1.09	0.5 (−3.4 to 4.3)	0.1 (−3.1 to 3.4)	−0.2 (−4.1 to 3.7)	2.8 (−1.1 to 6.7)	2.2 (−1.1 to 5.5)	2.7 (−1.2 to 6.6)
Hand Temperature	−0.7 (−7.2 to 5.8)	0.7 (−4.9 to 6.2)	0.8 (−5.7 to 7.4)	1.1 (−5.4 to 7.6)	0.4 (−5.1 to 6.0)	−0.001 (−6.6 to 6.5)	1.7 (−4.8 to 8.2)	−0.2 (−5.8 to 5.3)	−0.8 (−7.4 to 5.7)

**Table 4 jcm-11-01832-t004:** Subjective measure. Statistical test: ^(a)^ Pearson’s chi-squared test, ^(b)^ Fisher’s exact test, ^(c)^ One-way analysis of variance (one-way ANOVA).

*n* (%)	10 kHz (*n* = 20)	20 kHz (*n* = 20)	Sham Stimulation (*n* = 20)	*p* Value ^(a)(b)^
Pain sensation	0 (0%)	1 (5%)	0 (0%)	*p = 0.36*
Numbness	8 (40%)	4 (20%)	3 (15%)	*p = 0.15*
Cold sensation	8 (40%)	5 (25%)	0 (0%)	*p = 0.34*
Loss strength	8 (40%)	8 (40%)	2 (10%)	*p = 0.057*
Heaviness sensation	2 (10%)	5 (25%)	4 (20%)	*p = 0.37*
Tingle sensation	5 (25%)	1 (5%)	0 (0%)	*p = 0.02*
Inflammation	0 (0%)	0 (0%)	0 (0%)	*NA*
Erythema	0 (0%)	0 (0%)	0 (0%)	*NA*
Hot sensation	0 (0%)	0 (0%)	0 (0%)	*NA*
**Mean (SD)**	**10 kHz (*n* = 20)**	**20 kHz (*n* = 20)**	**Sham Stimulation (*n* = 20)**	***p* Value ^(c)^**
Pain intervention (0–10)	4.95 (1.9)	3.45 (1.9)	3.2 (1.8)	*0.19*
Unpleasant (0–10)	4.75 (2.1)	4.5 (2.0)	4.5 (1.6)	*0.87*

**Table 5 jcm-11-01832-t005:** Statistical analysis of blinding assessment. * Wishful thinking participants tend to think they are allocated to the active group even if not in reality.

Participants Results				
Methods	Index	*p*-value	95% confidence interval	Conclusion
James	0.44	0.098	0.37 to 0.51	Blinded
Bang − Active/2 × 5	0.8	0	0.72 to 0.87	Unblinded
Bang − Placebo/2 × 5	−0.72	1	−0.92 to −0.53	Opposite Guess *
**Assessor Results**				
Methods	Index	*p*-value	95% confidence interval	Conclusion
James	0.35	<0.001	0.24 to 0.45	Unblinded
Bang − Active/2 × 5	0.52	<0.001	0.36 to 0.69	Unblinded
Bang − Placebo/2 × 5	0.47	<0.001	0.25 to 0.70	Unblinded

## Data Availability

Original datasets are available in the Zenodo repository at DOI: https://doi.org/10.5281/zenodo.5840197 (accessed on 12 January 2022).

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
