# Peer review of "Effect of Percutaneous Electric Stimulation with High-Frequency Alternating Currents on the Sensory-Motor System of Healthy Volunteers: A Double-Blind Randomized Controlled Study"

_jcm, 2022, doi:10.3390/jcm11071832_

Round 1
Reviewer 1 Report
Thank you for the opportunity to review the manuscript. The manuscript has some points that need to be better clarified. Are they:
1.Objective needs to be better written. Describing the participating population.
2. First, the manuscript must be suitable for CONSORT. For further complete analysis of the manuscript.
3. What is the point of verifying the effect of percutaneous electrical stimulation with 10 kHz 57 and 20 kHz HFAC applied to the median nerve compared to sham stimulation on motor 58 response, somatosensory thresholds, and peripheral nerve conduction in healthy individuals?
4. Results, discussion and conclusion should be based on differences between groups. Mainly the conclusion.
Author Response
We would like to thank the reviewer for the constructive review, the positive comments and appreciation of our work. We believe that the proposed changes will make our manuscript more robust. The reviewer’s comments (in bold) and detailed responses to each point of concern are listed below. Changes in the new manuscript are marked up using the “Track Changes” function.
Thank you for the opportunity to review the manuscript. The manuscript has some points that need to be better clarified. Are they:
1.Objective needs to be better written. Describing the participating population.
Thank you for this suggestion. In the abstract section and in the title, we stated that our study was carried out in healthy participants, but this information was not added in the main aim.
In the new version of the manuscript, we have better defined the population in the objective, and rewritten the secondary objectives (Lines 57-60).
- First, the manuscript must be suitable for CONSORT. For further complete analysis of the manuscript.
We have added in section “2. Methods”, that our study was developed following CONSORT guidelines.
Line 62-63: “This study was performed according to the CONsolidated Standards of Reporting Trials (CONSORT) standards’ protocol.
In addition, according to your suggestion, we have completed the CONSORT 2010 checklist (see in attached file) and added in the manuscript the missing items.
- What is the point of verifying the effect of percutaneous electrical stimulation with 10 kHz and 20 kHz HFAC applied to the median nerve compared to sham stimulation on motor response, somatosensory thresholds, and peripheral nerve conduction in healthy individuals?
Until our knowledge, this is the first study that test this approach by stimulating the peripheral nerve percutaneously. As a previous step to test the effectiveness in a clinical population, we decided to investigate the effects of the stimulation and to evaluate the possible adverse events in non-injured subjects.
With the results of this study and with previous studies we have the needed evidence to implement this intervention in clinical studies with patients with specific pathologies. This information is included in the introduction and the discussion section.
- Results, discussion and conclusion should be based on differences between groups. Mainly the conclusion.
We agree with the reviewer that the key points are between groups differences. We have rewritten discussion and conclusion emphasizing on the differences observed between groups. Also, we have modified the abstract.
Below you can find the conclusions of the new version:
“The percutaneous ultrasound-guided application of HFAC at a frequency of 20 kHz to the median nerve produced a post-intervention reduction in strength and an increase in the myotonometry frequency when compared to sham group, that were rapidly reversible. However, no differences were found between 10 kHz and sham group”.

Reviewer 2 Report
Álvarez et al present the results of a small blinded study of 60 subjects to percutaneous high-frequency block of the median nerve on motor and sensory function. The manuscript is scientifically sound and the language sufficiently clear. The presentation of the results could be improved (I, for one, do not find walls of numbers presesented as tables to be particularly enlightening).
A few specific comments follow:
Section 2.6, regarding the post-hoc bonferonni correction. What p-value gives a p=0.05 family-wise error rate across all reported p-values in the manuscript?
Section 3.1, "Differences in the logarithmic decrement were observed" ... Please clarify. Is this the log of Myotonometry Decrement? if so, why was the comparison performed on the log values?
Table 2, would be very helpful to have baseline values and 95% CI in real units to give context to % change comparisons.
Table 3, typo suspected: is strength supposed to have 95% CI -19.8 to 1.8? Same with PPT. In general please check tables carefully to make sure 95% CIs are presented correctly.
Discussion, On the point regarding safety it is worth considering that this is a comparison of a single session vs a chronic many-session implant ... the safety profile of perc HFAC over many sessions may not be as favorable compared to an implant in the chronic case. I would be more conservative regarding the scare claims of implant repositioning (which is fairly rare)
Author Response
We would like to thank the reviewer for the constructive review, the positive comments and appreciation of our work. We believe that the proposed changes will make our manuscript more robust. The reviewer’s comments (in bold) and detailed responses to each point of concern are listed below. Changes in the new manuscript are marked up using the “Track Changes” function.
Reviewer: 2
Comments to Author
Álvarez et al present the results of a small, blinded study of 60 subjects to percutaneous high-frequency block of the median nerve on motor and sensory function. The manuscript is scientifically sound and the language sufficiently clear. The presentation of the results could be improved (I, for one, do not find walls of numbers presented as tables to be particularly enlightening).
A few specific comments follow:
Section 2.6, regarding the post-hoc Bonferroni correction. What p-value gives a p=0.05 family-wise error rate across all reported p-values in the manuscript?
In order to clarify the post-hoc Bonferroni, in the new version of the manuscript we have added in table 2 and 3 the exact p-values (not asterisks) in bold (those that are significant with p<0.05).
In addition, in section 2.6 we have clarified this point, specifying: “In the post-hoc analysis (Tables 2 and 3) significant p-values are shown at the nominal level alpha < 0.05.”
Section 3.1, "Differences in the logarithmic decrement were observed" ... Please clarify. Is this the log of Myotonometry Decrement? if so, why was the comparison performed on the log values?
We understand that myotonometry is a new tool to assess muscle tone and could be confusing. In order to clarify: the outcome Myotonometry has three variables to assess the mechanical muscle properties: Stiffness (N/m), Frequency of oscillation (Hz) and Decrement. Decrement is reported by the device in a logarithmic scale without units (absolute values) and this parameter serves to characterize tissue elasticity by measuring the loss of mechanical energy as the amplitude of the oscillations decreases. Although the explanation of the outcome has not been changed in the methods section, we have re-named in the results section the outcome “decrement parameter” in order to “logarithmic decrement” to avoid misunderstanding.
Table 2 would be very helpful to have baseline values and 95% CI in real units to give context to % change comparisons.
In line 212-213 we added a table with raw results in order to give context as you suggest.
“The raw results for all variables can be found in the supplementary material (Supplementary Appendix 1).”
Table 3, typo suspected: is strength supposed to have 95% CI -19.8 to 1.8? Same with PPT. In general, please check tables carefully to make sure 95% CIs are presented correctly.
Thank you for this appreciation. We have checked all tables and modified the typo in strength and PPT.
Discussion, On the point regarding safety it is worth considering that this is a comparison of a single session vs a chronic many-session implant ... the safety profile of perc HFAC over many sessions may not be as favorable compared to an implant in the chronic case. I would be more conservative regarding the scare claims of implant repositioning (which is fairly rare)
We agree with the reviewer at this point. It is needed to know the safety of chronic interventions with percutaneous HFAC approach. We have added in the new version of the manuscript a pair of sentences discussing this point (lines 306-308).

Round 2
Reviewer 1 Report
Thank you for your attention to clarifying the questions and suggestions requested in the last review. And even more, for the excellent work of reviewing the manuscript.